# Assessment of the histone mark-based epigenomic landscape in human myometrium at term pregnancy

**San Pin Steve Wu[1†], Elvis Quiroz[1†], Tianyuan Wang[2], Skylar G Montague Redecke[1], Xin Xu[3], Lin Lin[1,4], Matthew L Anderson[5], Francesco J DeMayo[1]\***

[1]Reproductive and Developmental Biology Laboratory, National Institue of Environmental Health Sciences, National Institutes of Health, Durham, United States; [2]Biostatistics and Computational Biology Branch, University of South Florida Morsani College of Medicine and Moffitt Cancer Center, Tampa, United States; [3]Epigenomic and DNA Sequencing Core Laboratory, National Institute of Environmental Health Sciences, National Institutes of Health, Durham, United States; [4]School of Nursing, University of California, San Francisco, San Francisco, United States; [5]Department of Obstetrics and Gynecology, University of South Florida Morsani College of Medicine and Tampa General Hospital Cancer Institute, Tampa, United States

**\*For correspondence:**
demayofj@niehs.nih.gov

†These authors contributed equally to this work

**Competing interest:** The authors declare that no competing interests exist.

## eLife Assessment

This **valuable** study employed a multi-omics approach to elucidate the regulatory mechanism underlying parturition and myometrial quiescence. The data presented to support the main conclusion are **solid**. This work will be of interest to both basic researchers who work on reproductive biology and clinicians who practice reproductive medicine.

**Abstract** The myometrium plays a critical role during pregnancy as it is responsible for both the structural integrity of the uterus and force generation at term. Emerging studies in mice indicate a dynamic change of the myometrial epigenome and transcriptome during pregnancy to ready the contractile machinery for parturition. However, the regulatory systems underlying myometrial gene expression patterns throughout gestation remain largely unknown. Here, we investigated human term pregnant nonlabor myometrial biopsies for transcriptome, enhancer histone mark cistrome, and chromatin conformation pattern mapping. More than thirty thousand putative enhancers with H3K27ac and H3K4me1 double positive marks were identified in the myometrium. Enriched transcription factor binding motifs include known myometrial regulators AP-1, STAT, NFkB, and PGR among others. Putative myometrial super enhancers are mostly colocalized with progesterone receptor-occupying sites and preferentially associated with highly expressing genes, suggesting a conserved role of PGR in regulating the myometrial transcriptome between species. In human myometrial specimens, inferred PGR activities are positively correlated with phospholipase C like 2 (*PLCL2*) mRNA levels, supporting that PGR may act through this genomic region to promote *PLCL2* expression. PGR overexpression facilitated *PLCL2* gene expression in myometrial cells. Using CRISPR activation, we assessed the functionality of a PGR putative enhancer 35 kilobases upstream of the contractile-restrictive gene *PLCL2*. In summary, the results of this study serve as a resource to study gene regulatory mechanisms in the human myometrium at the term pregnancy stage for further advancing women's health research.

## Introduction

The myometrium is the muscular component of the uterus. Throughout gestation, the myometrium maintains contractile quiescence until labor, where uterine contractility facilitates parturition. Pre-term birth, often characterized by aberrant myometrial activity before term, results in the termination of pregnancy prior to 37 weeks gestation. Complications associated with pre-term birth account for 35% of neonatal death and contribute to lifelong physical and socioeconomic multi-morbidities to surviving neonates and mothers (*Chawanpaiboon et al., 2019*). It is therefore critical to understand the molecular mechanisms underlying the maintenance and shift of myometrial quiescence throughout gestation.

The myometrium undergoes extensive structural and functional remodeling in preparation for parturition through genomic regulatory mechanisms which influence gene expression throughout pregnancy. Major transcription factor families have been identified to contribute throughout the remodeling process. Studies on the role of steroid hormone receptors in myometrial remodeling suggest that the withdrawal of functional progesterone signaling at term results from a stoichiometric shift favoring the PGR-A isoform over PGR-B. This shift is associated with increased activation of estrogen receptor alpha (ESR1) expression at term (*Mesiano et al., 2002*; *Merlino et al., 2007*; *Muter et al., 2016*).

Activator protein 1 (AP-1) complex subunits have been observed to act as PGR coregulators (*Dai et al., 2003*) and have dynamic expression patterns throughout gestation in both humans and rodent models. For example, FOS:JUN heterodimers are implicated to be critical for the initiation of labor through transcriptional regulation of gap junction proteins such as Gja1 (Gap junction alpha 1; *Mitchell and Lye, 2001*; *Mitchell and Lye, 2005*; *Balducci et al., 1993*).

Contributing to the dynamic nature of the myometrial transcriptome during term is the epigenome and its reprogramming. Studies investigating epigenetic markers related to gene activation in the mouse myometrium have revealed that the promoters for contractility-driving genes, such as Gja1, are epigenetically activated well before the onset of labor (*Shchuka et al., 2020*). Considering this, the myometrial epigenome's role in parturition disorders, such as preterm birth, has thus been under investigation. For example, it is known that altered DNA methylation is linked with preterm birth (*Barcelona de Mendoza et al., 2017*), including at the promoters of genes associated with myometrial contraction (*Erickson et al., 2023*) and fetal membrane rupture (*Wang et al., 2008*). However, differential myometrial DNA methylation at CpG islands in the promoters of contractility-driving genes is not thought to be a major contributor to preterm birth (*Mitsuya et al., 2014*). Given that DNA methylation-mediated gene regulation often occurs outside of CpG islands (*Irizarry et al., 2009*), there is still work to be done at this interface. Regardless, whether through epigenetic or transcriptomic gene regulatory programs, the molecular mechanisms underlying the regulation of genes critical for myometrial reprogramming and quiescent maintenance are still poorly understood.

In this study, we aim to improve our understanding of the cellular processes involved in shaping the myometrium's dynamic state of quiescence at pregnancy through the epigenetic and transcriptomic profiling of the human myometrium. Because of the relevance of genomic regulatory mechanisms in coordinating myometrial activity, we identified candidate cis-acting regulatory regions using chromatin immunoprecipitation sequencing (ChIP-Seq) assays of surrogate histone enhancer markers H3K27ac (*Creyghton et al., 2010*) and H3K4me1 (*Heintzman et al., 2007*) alongside chromatin conformation capture assays in term pregnant myometrial tissues. Using this data, we investigated cis-acting regulatory regions for the gene phospholipase C like 2, which encodes for the protein PLCL2 and has been implicated in the modulation of calcium signaling (*Uji et al., 2002*) and maintenance of myometrial quiescence (*Peavey et al., 2021*). Putative enhancers upstream of the PLCL2 transcriptional start site were subjected to functional assessment using CRISPR activation-based assays. Here, we identified a genomic region 35 kilobases upstream of the *PLCL2* transcriptional start site involved in *PLCL2* transcriptional regulation. Furthermore, we aimed to characterize this genomic region through the identification of candidate *PLCL2* transcriptional regulators co-localizing with this cis-acting element using integrative cistrome and transcriptome analysis and have demonstrated PGR to be a direct regulator for *PLCL2* expression. These findings build upon our understanding of myometrial remodeling throughout gestation and will be pertinent for the development of medical interventions aiming to address pre-term birth.

**Table 1.** Epigenome and transcriptome profile of the individual myometrium biopsy. Active genes are defined as FPKM ≥1. N.D., not determined.

| Subject | 1 | 2 | 3 |
|---|---|---|---|
| Number of H3K27ac intervals | 47,223 | 39,465 | 46,026 |
| Number of H3K4me1 intervals | 72,091 | 74,511 | 76,374 |
| Number of active genes | 12,809 | 11,761 | 11,902 |
| Number of chromatin loops | 10,321 | N.D. | 16,841 |

## Results

### Epigenomic landscape in term pregnant myometrial specimens

To better understand the regulatory network shaping the myometrial transcriptome before labor, we analyzed transcriptome and putative enhancers in individual human myometrial specimens. Using RNA-seq, we identified actively expressed RNAs, while ChIP-seq for H3K27ac and H3K4me1 was used to map putative enhancers (*Supplementary file 1*). Active genes were associated with nearby putative enhancers based on their genomic proximity. Additionally, chromatin looping patterns were mapped using Hi-C to further link active genes and putative enhancers within the same chromatin loops (*Supplementary file 1* and *Supplementary file 2*). The ChIP-seq assay identified an average of 44,238 H3K27ac and 74,325 H3K4me1-positive genomic regions in three term-pregnant, nonlabor myometrial biopsies (*Table 1*). An RNA-Seq assay revealed an average of 12,157 active genes in these specimens that manifested expression levels of fragments per kilobase of transcript per million mapped reads (FPKM) greater than or equal to 1 (*Table 1*). A High-throughput Chromosome Conformation Capture (Hi-C) assay further found a total of 27,162 chromatin loops from two of the three myometrial specimens (*Table 1*, subject 1 and 3). We failed to identify chromatin loops in the second subject's biopsy due to limited sample availability. Together, these results delineate a map of H3K27ac and H3K4me1-positive signals in the human term pregnant myometrial tissue before the onset of labor, which we use as a resource to investigate the molecular mechanisms that prepare the myometrium for subsequent parturition.

When comparing the present study to previous findings (*Dotts et al., 2023*), 8563 genomic regions carry common H3K27ac-positive histone marks. However, significant variations in the number and location of H3K27ac-positive intervals are present across the six samples among these two studies (*Figure 1—figure supplement 1*). Depending on the number of mapped intervals, the 8563 common regions constitute between 20.7% and 71.5% of total H3K27ac-positive intervals across the six myometrial specimens (*Figure 1—figure supplement 1*). Notably, the fraction of intervals commonly identified in the present study's three specimens ranges between 56.4% and 70.0% of each of their total H3K27ac-positive regions, while the Dotts dataset has a wider range between 28.0% and 74.0%. With a less stringent criterium, most of the H3K27ac-positive intervals are found in at least two samples, ranging between 75.4% and 98.3% (*Figure 1—figure supplement 1*). These data together highlight the degree of variation in mapping the epigenome among specimens and datasets.

### Putative enhancers for gene regulation in the human myometrium

Since H3K27ac and H3K4me1 marked genome regions are associated with enhancers for gene regulation (*Creyghton et al., 2010*; *Heintzman et al., 2007*), we define the regions that have overlapping H3K27ac and H3K4me1 marks as putative myometrial enhancers at the term pregnant nonlabor stage (*Supplementary file 3*). Among the three specimens, 13,114 putative enhancers are commonly present (*Figure 1A*). A significant variance in the location of myometrial putative enhancers is also seen among biopsies as evident by the observation that these common putative enhancers make up less than half of the total putative enhancers in each individual specimen (*Figure 1A*). More than one-third of these 13,114 common putative enhancers are located within a 100-kilobase vicinity of actively high-expressing genes (FPKM≥15), while a much smaller fraction of them (13.3%) are associated with actively low-expressing genes (*Figure 1B*). Motif enrichment analysis on these common putative enhancers further reveals an overrepresentation of binding motifs for myometrial transcription factors AP-1, STAT5, and NFκB, steroid hormone receptors PGR and NR3C1, and smooth muscle

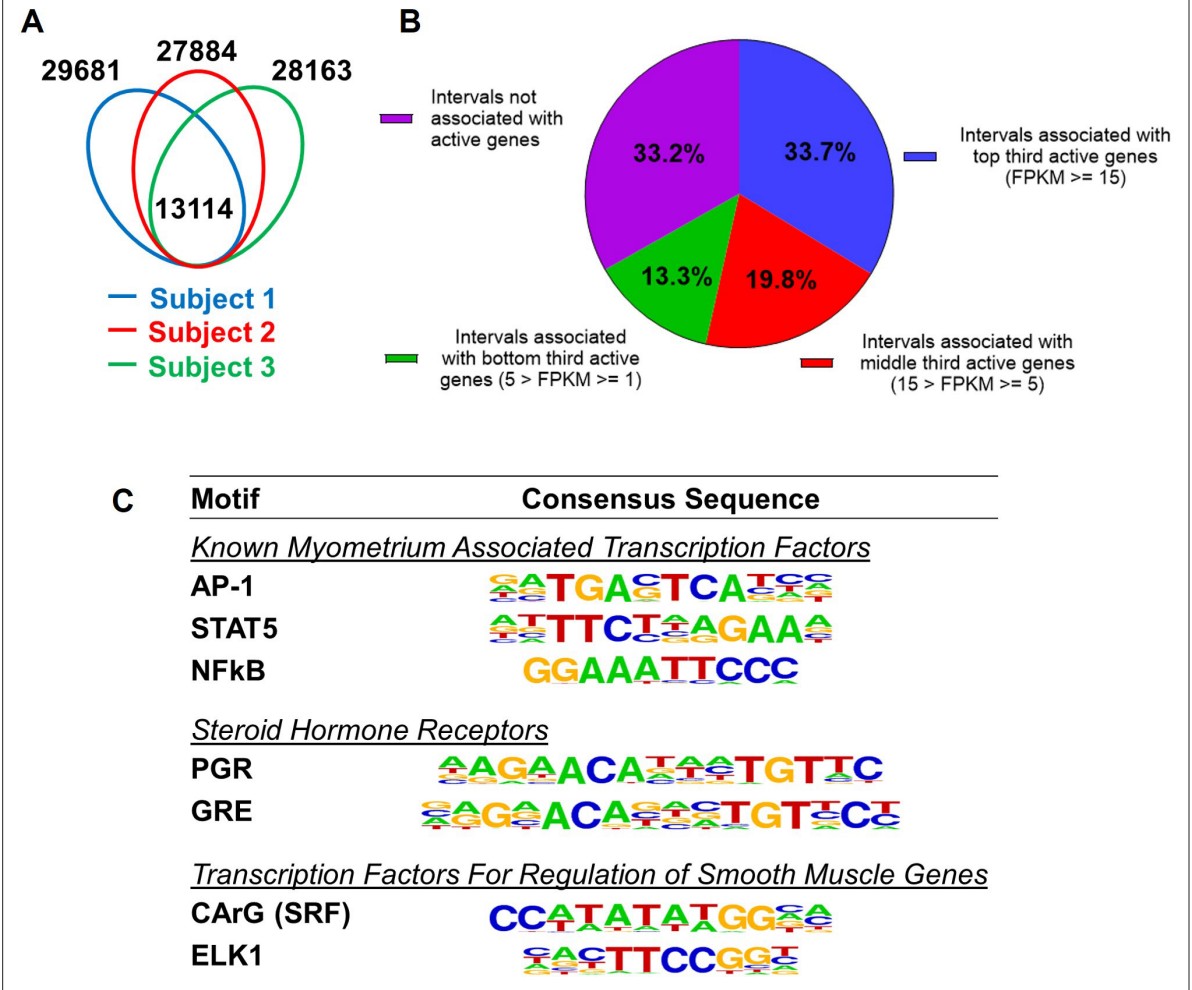

**Figure 1.** Putative enhancers in term pregnant human myometrial tissues. (**A**) Distinct and common putative enhancers in term pregnant myometrial biopsies from three subjects. Genomic regions with H3K27ac and H3K4me1 double positive histone marks are defined as putative active enhancers. (**B**) Association of commonly shared putative enhancers with active genes. The association between an interval and an active gene is defined by locating within 100 kb vicinity of each other. (**C**) Over-Represented transcription factor binding motifs in putative enhancers. A subset of enriched motifs that are relevant to myometrial homeostasis in the 13,090 H3K4me1/H3K27ac-positive putative enhancer regions is displayed.

The online version of this article includes the following figure supplement(s) for figure 1:

**Figure supplement 1.** Subject variations on H3K27ac-positive histone marks.

transcription regulators SRF and ELK1 (*Figure 1C*, *Supplementary file 4*). These findings collectively suggest that the putative myometrial enhancers bring together smooth muscle and hormonal control programs for the regulation of myometrial gene expression at term pregnancy.

Super enhancers house hormone-dependent gene regulatory programs for female reproductive tract homeostasis (*Shin et al., 2016*; *Hewitt et al., 2020*). Across the three human subjects, 540 putative super enhancers are commonly identified in the term pregnant nonlabor myometrial specimens (*Figure 2A*, *Supplementary file 5*). More than 40% of the 540 putative super enhancers are located within a 100 kilobase distance to high-expressing genes (FPKM ≥ 15), while only 7.3% of putative myometrial super enhancers are found near low-expressing genes (5>FPKM ≥ 1; *Figure 2B*). Compared with the regular putative enhancers, the putative myometrial super enhancers are found more frequently near active genes that are expressed at relatively higher levels (*Figures 1B and 2B*). Notably, 76% of the putative super-enhancers co-localize with known PGR-occupied regions in human myometrial tissue, compared to 20% co-localization observed in regular enhancers. Further examining the myometrial active genes that are associated with putative super enhancers revealed an enrichment of gene functions in cytoskeleton organization, extracellular-receptor interaction, transcription

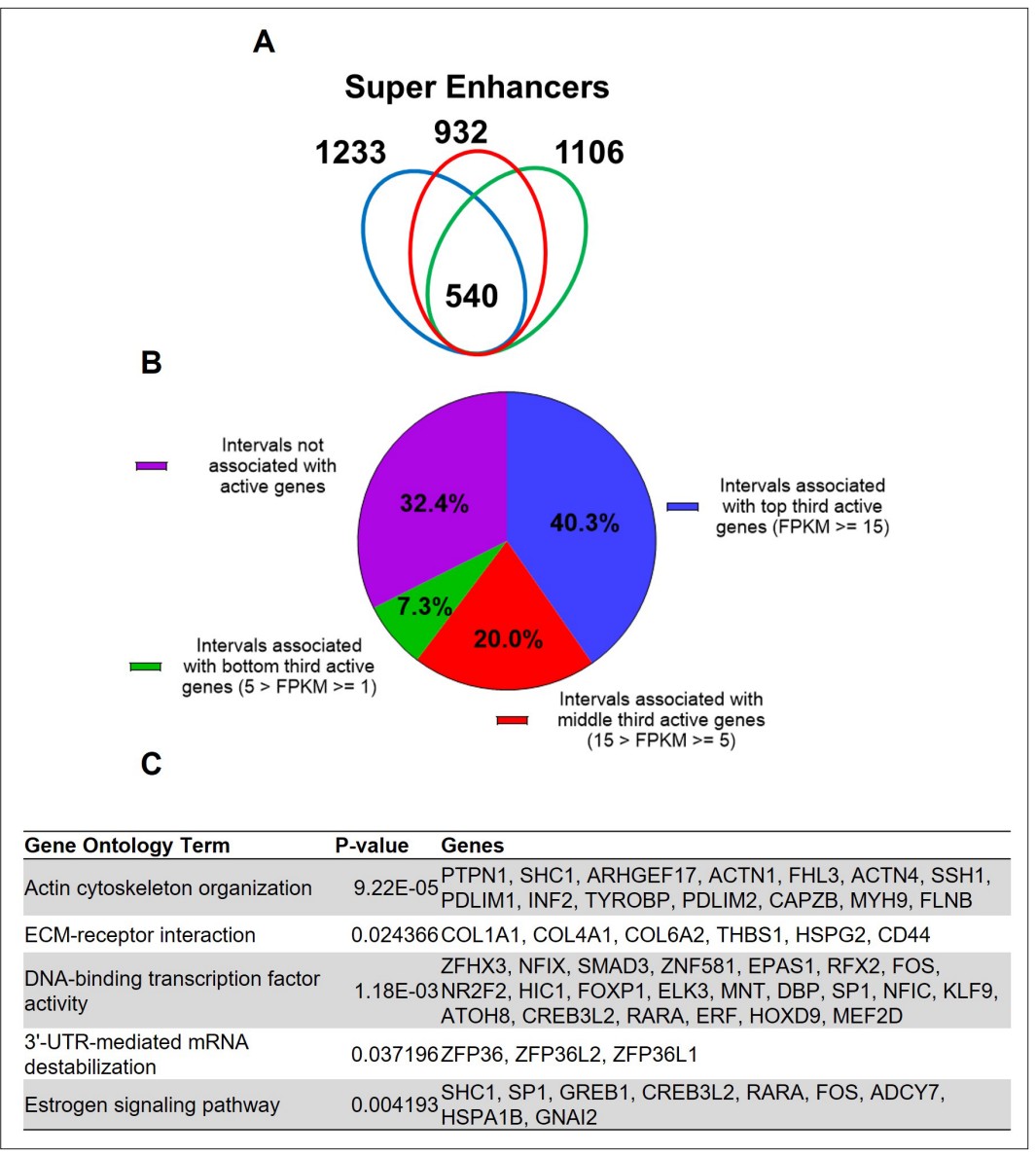

**Figure 2.** Putative super enhancers in term pregnant human myometrial tissues. (**A**) Number of super enhancers mapped in tissues of each individual human subject. (**B**) Association of commonly shared super enhancers with active genes in human myometrium. (**C**) Selected enriched gene ontology terms in the 346 active genes that are associated with super enhancers.

The online version of this article includes the following figure supplement(s) for figure 2:

**Figure supplement 1.** PGR occupancy in myometrial enhancers.

regulation, mRNA stability, and estrogen signaling (*Figure 2C*, *Supplementary file 6*). Taken together, these results establish the association among the putative myometrial enhancers, the potential regulatory program within these super enhancers, and the cellular functions of the genes they may control.

## Cis-acting elements for the control of the contractile gene *PLCL2*

We previously demonstrated the positive correlation of PLCL2 and PGR expression in a mouse model and PLCL2's function on negatively modulating oxytocin-induced myometrial cell contraction (*Peavey et al., 2021*). However, the mechanism that underlies PGR's regulation of PLCL2 remains unclear. Taking advantage of the mapped myometrial cis-acting elements, we aimed to identify the cis-acting

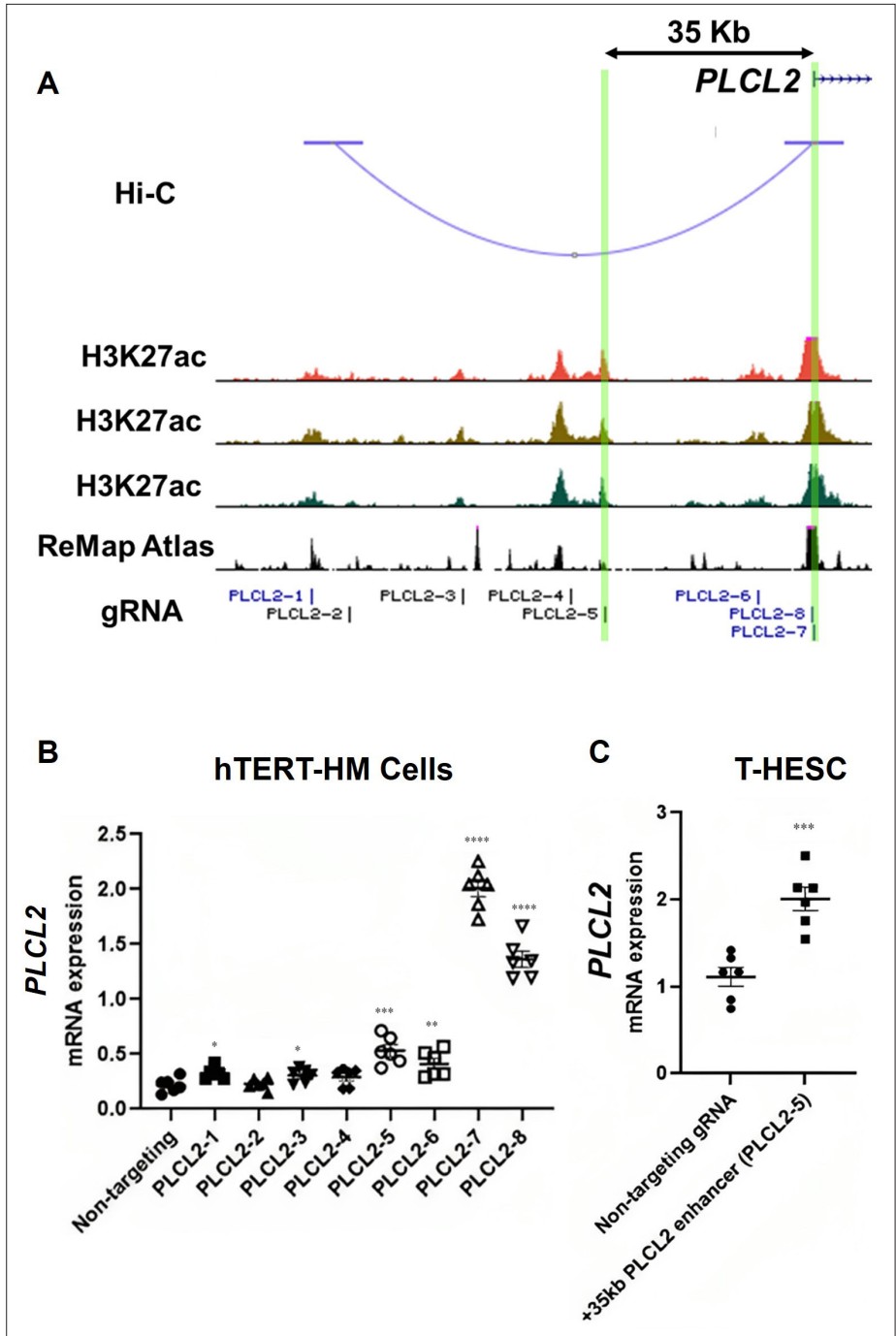

**Figure 3.** Identification of enhancers for the *PLCL2* gene. (**A**) UCSC Genome Browser track view of the human PLCL2 locus marked with gRNA targeting locations. (**B, C**) Relative PLCL2 mRNA levels measured by qRT-PCR in hTERT-HM cells (**B**) or T-HESC cells (**C**) that express denoted gRNAs with the CRISPR activator (N=3 with technical duplicates). Statistical significance was determined using unpaired t-test for comparisons between two groups, or one-way ANOVA for comparisons between >2 groups. Significance levels are denoted as follows: ****, $p<0.0001$; ***, $p<0.001$; **, $p<0.01$; *, $p<0.05$. Error bars are shown as mean with SEM.

elements that may contribute to the PLCL2 transcriptional regulation with a special interest in the PGR-related enhancers.

The Hi-C assay identified a chromatin loop with one end at the *PLCL2* transcription start site (TSS) and the other end at approximately 90 kilobases upstream of the TSS (*Figure 3A*). Within this chromatin loop, ChIP-Seq results revealed seven genomic regions that are marked with H3K27ac in all

three myometrial specimens (*Figure 3A*). Six of the seven regions are also co-localized with previously published genome occupancy of transcription regulators curated by the ReMap Atlas (*Hammal et al., 2022*; *Figure 3A*). For the purpose of functional screening, we focus on H3K27ac signals instead of using the H3K27ac/H3K4me1 double positive criterium to cast a wider net. Eight guide RNAs (gRNAs) were designed to target these seven candidate regions for CRISPR activation (CRISPRa)-based screening of *PLCL2* gene regulation (*Figure 3A*). The two previously reported gRNAs (PLCL2-7 and PLCL2-8) were able to elevate the endogenous *PLCL2* mRNA levels significantly higher than the non-targeting gRNA in immortalized human myometrial cells (hTERT-HM; *Figure 3B*; *Peavey et al., 2021*). gRNAs PLCL2-1, PLCL2-3, PLCL2-5, and PLCL2-6 also significantly increased *PLCL2* expression above the levels of the non-targeting group (*Figure 3B*). Moreover, results from the Perturb-Seq assay found that the PLCL2 gene ranks high in the differentially expressed gene list of the PLCL2-5 gRNA expressing cells compared with the non-targeting gRNA expressing cells (*Supplementary file 7* and *Supplementary file 8*), in line with the qRT-PCR assay finding in which the gRNA PLCL2-5 induced *PLCL2* expression to the greatest extent among the upstream non-coding genome targeting gRNA (*Figure 3B*). Moreover, gRNA PLCL2-5 was capable of mediating the increase of the *PLCL2* transcript abundance in another uterine mesenchymal cell type T-HESC (*Figure 3C*). Collectively, these findings support the PLCL2-5-targeted genomic region as a cis-acting element for regulation of the *PLCL2* gene.

## PGR as an upstream regulator for the *PLCL2* gene in the human myometrium

After determining its cis-acting element role, the PLCL2-5 targeted genomic region was further examined to identify upstream regulators that control myometrial *PLCL2* expression. This was achieved by first identifying transcription factor occupancy in this genome region in any tissues/cells that are documented in public databases (ReMAP Atlas; *Hammal et al., 2022*), followed by filtering for factors that have known mRNA and/or protein expression in hTERT-HM cells and tissues using publicly available transcriptomic and histological data including the Human Protein Atlas (Human Protein Atlas proteinatlas.org; *Uhlén et al., 2015*) and published RNA-Seq datasets (*Stanfield et al., 2019*; *Wu et al., 2020*). The resulting candidate upstream regulators for the myometrial *PLCL2* gene are documented in *Table 2*. Since the *PLCL2* mRNA abundance is much lower in the hTERT-HM cells than in the myometrial tissues, these candidate upstream regulators were further grouped into candidate activators and repressors based on an assumption that a candidate activator would be expressed higher in human myometrial tissue, and a candidate repressor being expressed higher instead in the hTERT-HM cells, holding the *PLCL2* mRNA levels down. Candidate activators for the *PLCL2* gene include known myometrial regulators such as PGR, estrogen receptor alpha (ESR1), and AP-1 (*Table 2*). Interestingly, the mediator complex protein subunit MED1, cohesion complex members RAD21 and SMC3, and many epigenomic modifiers are among the list of the candidate repressors (*Table 2*). These findings not only provide candidates for subsequent functional assessment, but also highlight potential pathways for future investigations on the regulation of the contractile gene *PLCL2* expression in the myometrium.

We previously demonstrated the regulation of mouse *Plcl2* gene by the myometrial PGR (*Wu et al., 2022*). To further test whether such a regulatory mechanism is also present in humans, the CRISPRa technology was employed to increase the expression of PGR mRNA and proteins in hTERT-HM cells (*Figure 4A and B*). Given that the PGR640 gRNA led to higher PGR protein production (*Figure 4B*), this gRNA was utilized to stimulate myometrial *PGR* expression for subsequent experiments. Indeed, RT-qPCR results showed that PGR overexpression increased *PLCL2* mRNA abundance in hTERT-HM cells (*Figure 4C*), demonstrating a consistent finding with the mouse model.

To further explore this regulatory relationship in the human myometrial tissue, we examined the degree of correlation between the inferred PGR activity and the *PLCL2* transcript abundance in myometrial biopsies. Gene expression profiles of myometrial specimens from 13 proliferative-phase, 6 secretory-phase, 14 post-menopausal, and 10 Provera-treated human subjects were determined by the RNA-Seq assay (*Supplementary file 9*) to take advantage of the wide spectrum of progesterone signaling activities in these samples. Inferred myometrial PGR activities, represented as T-scores (*Supplementary file 9*), were derived from this normalized gene expression matrix of 43 human myometrial specimens with the previously published mouse myometrial *Pgr* gene signature (*Wu et al.,*

**Table 2.** Candidate upstream regulators that mediate PLCL2-5 enhancer's regulatory effect on PLCL2 expression.

| Candidate Activators | Description |
| --- | --- |
| PGR | |
| ESR1 | Steroid Hormone Receptor |
| FOS, JUN | |
| JUN | |
| JUNB | |
| JUND | AP-1 Transcription Factor Subunit, bZIP Transcription Factor |
| MAF | bZIP Transcription Factor |
| BHLHE40 | bHLH Transcription Factor, Involved in CLOCK gene regulation |
| TCF21 | bHLH Transcription Factor |
| MAX | bHLHZ Transcription Factor |
| FOXO1 | Forkhead Box Transcription Factor |
| MRTFB | Myocardin Family, SRF Transcriptional Co-Activator |
| STAT3 | Transcriptional Regulator |
| ERG | ETS Transcriptional Regulator |
| **Candidate Repressors** | **Description** |
| ARID1a | SWI/SNF Family of Epigenetic Modifiers |
| BRD2 | |
| BRD4 | Epigenetic Modifier, Chromatin Reader |
| KMT2A | Component of MLL Epigenetic Modification Complex |
| SMARCb1 | Component of BAF Epigenetic Modification Complex |
| ZMYM3 | Component of Epigenetic Modification Complex |
| CREB1 | bZip Transcription Factor. Interacts with Epigenetic Modifiers |
| CREBP | CREB Binding Protein, Epigenetic Modifier |
| HCAC2 | |
| HDAC3 | Histone Deacetylase, Epigenetic Modifier |
| CLOCK | bHLH Transcription Factor Family, Rhythmic Epigenetic Modifier |
| MED1 | Mediator Complex Subunit |
| RELB | NF-KB Transcription Factor Subunit |
| CHD4 | Component of NuRD Epigenetic Modification Complex |
| RAD21 | |
| SMC2 | Cohesin Complex Member |
| REST | KLF Silencing Transcription Factor |
| TCF4 | bHLH Transcription Factor |
| ZNF687 | Zinc Finger Protein, Transcriptional Regulator |

*2022*; *Li et al., 2021*). A Pearson correlation analysis found a positive correlation ($r$=0.47) between the T-scores and normalized *PLCL2* mRNA levels in these 43 specimens (*Figure 4D*). Together, these in vitro and in vivo findings collectively support that PGR is a *PLCL2* activator, likely acting through the 35 kb upstream cis-acting element.

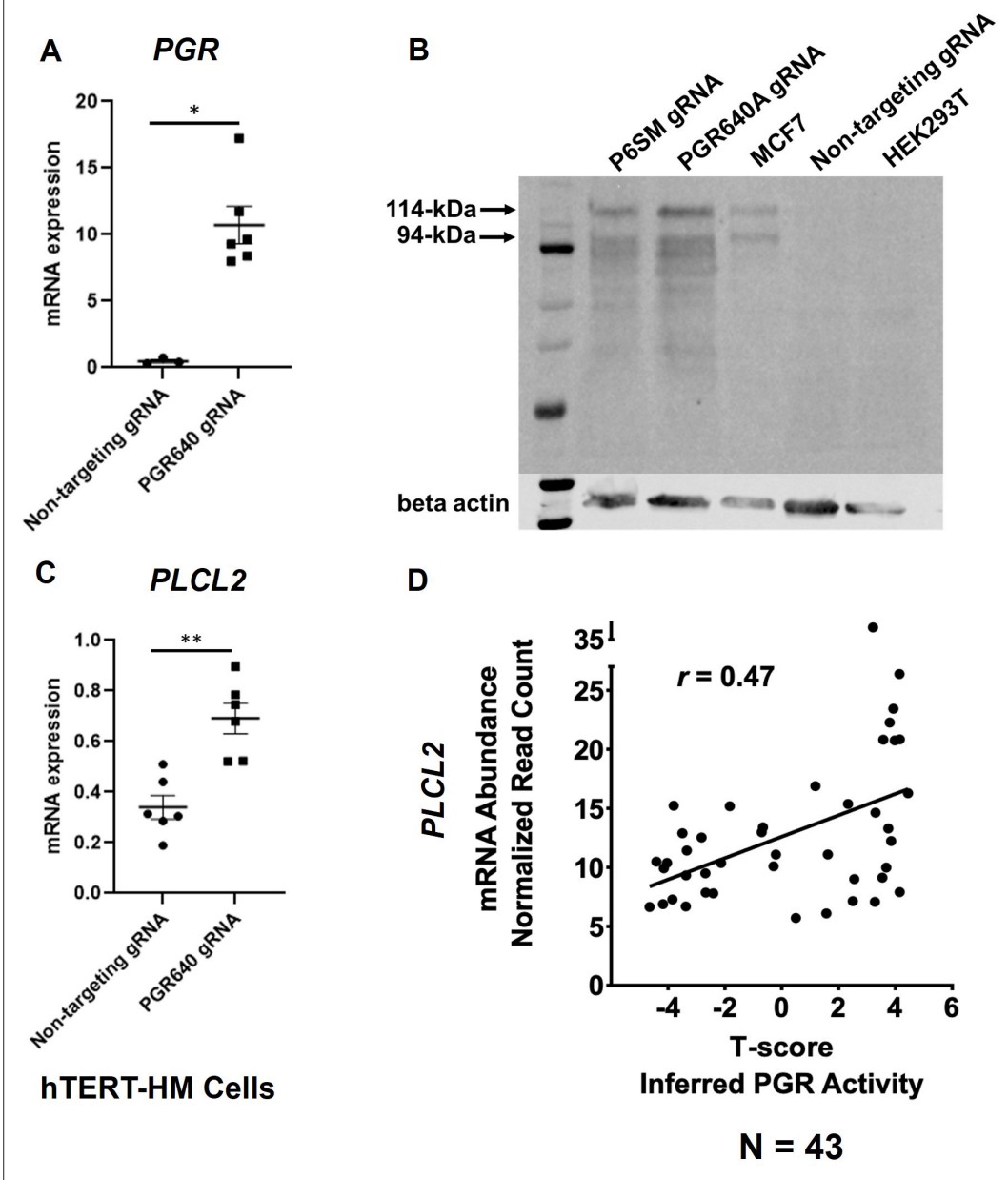

**Figure 4.** PGR regulation of *PLCL2* expression. (**A**) *PGR* mRNA abundance in hTERT-HM cells with the PGR-promoter targeting (PGR640) or non-targeting gRNAs under the CRISPR activation system. (N=3 with technical qPCR duplicates). (**B**) Protein abundance in hTERT-HM cells with the PGR-promoter targeting (PGR640 and P6SM) or non-targeting gRNAs under the CRISPR activation system. Protein extracts from unmanipulated MCF7 and HEK293T cells serve as positive and negative control for the PGR presence. (**C**) *PLCL2* mRNA abundance in hTERT-HM cells with the PGR-promoter targeting (PGR640) or non-targeting gRNAs under the CRISPR activation system. (N=3 with technical qPCR duplicates). (**D**) Pearson correlation between *PLCL2* mRNA levels and inferred PGR activities (T-Scores) in 43 human myometrial specimens. **, p<0.01; *, p<0.05 by Mann-Whitney test (**A**) and unpaired t-test (**C**). Error bars are shown as mean with SEM.

The online version of this article includes the following source data for figure 4:

**Source data 1.** Western blot analysis of GAPDH and PGR.

**Source data 2.** Western blot analysis of GAPDH and PGR, labelled.

## Discussion

Temporally and spatially regulated gene expression is the foundation of tissue phenotype manifestation. Shchuka and colleagues utilize the mouse model to show that the myometrial epigenome is readily programed for parturition-associated gene induction days ahead of the term pregnancy (*Shchuka et al., 2020*), laying the ground for subsequent laboring events. We chose to study the human myometrium at the term pregnant nonlabor stage in order to examine the association between the epigenome landscape and the gene expression pattern not only for preparing the myometrium for parturition, but also for maintaining myometrial quiescence. We also performed a multi-omic study on a myometrial specimen from each individual human subject to better align all the datasets together, reflecting specimen and subject individuality. While the variation among specimens is substantial, we were able to identify myometrial cis-acting elements commonly shared among three individuals. As a proof of principle, we subsequently used this information to functionally screen for genomic regions that may mediate the regulation of the contractile-suppressing gene *PLCL2.* The H3K27ac marked 35-kilobase upstream region as a result of the screening then serves as a bridge to identify PGR as one of the upstream regulators for modulating *PLCL2* gene expression in human myometrium. These findings showcase the value of these datasets as a resource for future investigations on the mechanisms of myometrial gene regulation.

Variations across cistromic datasets for the human myometrium are significant (*Figure 1*; *Figure 1—figure supplement 1*; *Dotts et al., 2023*). Contributing factors likely include but are not limited to subject-to-subject differences and batch effects from sample preparations. Taking the H3K27ac marked genomic regions as an example, regions commonly identified in the specimens of this study's three subjects account for 55.9%, 67.7%, and 57.9% of total H3K27ac-positive intervals in each individual sample, despite our best attempt to reduce the batch effect. Heterogeneity in cell type compositions, likely due to the sampling location, could be a major contributing factor to the variation. It is also speculated that the genomic regions with subject-to-subject variances could be transiently present, and thus, the signal could not be captured across samples. In addition, the underlying health conditions, medications, and environmental challenges among others may also affect the epigenomic profile. Variations could also arise from differences in the detection threshold. We identified an average of 44,238 H3K27ac marked regions per specimen, while Dotts and colleagues found nearly 19,000 per sample (*Dotts et al., 2023*). Reagent and sample handling could confound with biological variances of specimens, leading to this difference. Future investigations using standardized sample preparation protocols at the single-cell resolution may help to reduce this variation.

Results from the CRISPRa-based Perturb-Seq assay demonstrate the capability of this technology to screen for cis-acting elements that are in topological vicinity of the gene of interest. This assay is particularly useful on genes that are expressed at low levels in the screening platform, that is the cells. The dCas9VPR activator can generate satisfactory signal-to-noise ratios to be detected by single-cell RNAseq. The advantage of assaying with this system is that each individual cell serves as a container to generate data points instead of relying on multiple wells on cell culture plates. Multiplexing with multiple gRNAs can be achieved by using the gRNA sequence as a unique barcode of the cell, enabling the simultaneous collection of numerous data points for each individual gRNA. In the present study, 56.7% of tested cells carried one species of detectable gRNAs, permitting the study of the resulting effect under a single gRNA. Moreover, another 27.0% of assayed cells have more than one species of detectable gRNAs, which opens the possibility to study interactions between different cis-acting elements. However, we failed to use this approach for the purpose of identifying target genes of a putative enhancer of interest (*Supplementary file 4*). Challenges such as difficulties in defining the criteria of the associated genes and unsatisfactory signal-to-noise ratios on genes already at high baseline levels should be considered in future experimental designs. Taken together, the results of the present study support the use of the CRISPRa-based Pertub-Seq assay under a defined scope.

PLCL2 and its paralog PLCL1 are catalytically inactive members of the phospholipase C family functioning as negative regulators of calcium signaling (*Kanematsu et al., 2005*; *Takenaka et al., 2003*; *Uji et al., 2002*). Both of them are expressed in human myometrial tissues and are able to attenuate oxytocin-induced muscle cell contraction (*Peavey et al., 2021*). *PLCL1* mRNA abundance is higher in the myometrium at term pregnancy than the non-gravid stage, while *PLCL2* transcript levels remain comparable between the two stages (*Wu et al., 2020*). Data from cultured human endometrial stromal cells and the genetically engineered mouse model reveal progesterone signaling as the

upstream regulator of *PLCL1* and *PLCL2*, respectively (**Muter et al., 2016**; **Peavey et al., 2021**). The present study further demonstrates the PGR-dependent regulation of the *PLCL2* gene in the human myometrial cells, suggesting that this pathway is conserved between human and mouse. Importantly, the identification of the cis-acting element 35-kilobase upstream of the *PLCL2* gene opens an avenue to investigate the impact of interactions between PGR and other myometrial transcription regulators on the mechanism of action that controls the myometrial contractile machinery. The future findings in this space could provide insight on progesterone signaling modification and identify progesterone signaling modifiers for the development of novel therapy that targets parturition disorders.

## Materials and methods
### Collection of myometrial specimens
Permission to collect human tissue specimens was prospectively obtained from individuals undergoing hysterectomy or cesarean section for benign clinical indications (H-33461). Gravid myometrial tissue was obtained from the margin of the hysterotomy in women undergoing term cesarean sections (>38 weeks estimated gestational age) without evidence of labor. Non-gravid myometrial tissue was collected from pre-menopausal women undergoing hysterectomy for benign conditions. Specimens from gravid women receiving treatment for pre-eclampsia, eclampsia, pregnancy-related hypertension, or pre-term labor were excluded.

### Human cell lines
Human immortalized myometrial cells (hTERT-HM) (**Condon et al., 2002**) cells and human immortalized endometrial cells (T-hESC; ATCC, CRL-4003) were cultured in DMEM/F-12 (Invitrogen, Grand Island, NY, USA; Cat. No. 11320033.) supplemented with 10% Fetal Bovine Serum (FBS, Gibco; Cat. No. 26140079) and antibiotics (10 000 IU/mL penicillin, 10 000 IU/ mL streptomycin; Life Technologies, Grand Island, NY). Cell culture media was filtered using the 0.22 µm Rapid-Flow Sterile Disposable Filter Units (Nalgene). Cells were incubated at 5% $CO_2$ and 37 °C.

### RNA isolation and RT-qPCR
RNA was isolated from cells using TRIzol Reagent (Invitrogen; Cat. No. 15596026) and the RNeasy mini RNA isolation kit (QIAGEN, Hilden, Germany). cDNA was prepared using Moloney Murine Leukemia Virus reverse transcriptase (Thermo Fisher Scientific; Cat. No. 28025013) with Random Hexamers (Invitrogen, Waltham, MA, USA) according to manufacturer protocol. For quantitative analysis of mRNA, SsoAdvanced Universal SYBR Green Supermix (Bio-Rad, Hercules, CA, USA; Cat. No. 1725271) was used according to manufacturer instructions. Relative mRNA expression was determined by standard curve-based method (**Larionov et al., 2005**). Relative expression of genes of interest was normalized to the 18 S rRNA. Briefly, reaction samples were prepared to a volume of 20 µl with 5 uM of both forward and reverse primers, cDNA, and a final 1 x concentration of the SYBR Green Supermix. The following primers were used: PGR forward primer: TTTAAGAGGGCAATGGAAGG; PGR reverse primer: CGGATTTTATCAACGATGCAG; PLCL2 forward primer: TATGACATGATGATTCAGTCCCTC; PLCL2 reverse primer: TTCCTTGGTGCCTATGCTGT; 18 S forward primer: GTAACCCGTTGAACCC CATT; 18 S reverse primer: CCATCCAATCGGTAGTAGCG. The reaction was heated to 95 °C for 30 s, followed by 39 cycles of denaturation at 95 °C for 15 s and annealing and elongation at 60 °C for 30 s. Temperature cycles were performed on the CFX Connect Real-Time PCR Detection System (Bio-Rad). Experiments were performed in three biological replicates and two technical replicates, where biological replicates describe independent samples, and technical replicates describe repeated measurements on the same biological samples.

### Western blot assay
Protein was isolated from cells using the RIPA Lysis and Extraction Buffer (Thermo Scientific; Cat. No. 89900) with the following specifications: approximately 300,000 cells were pelleted and lysed with 100 µl of complete Pierce RIPA buffer. Protein was quantified using the BCA Protein Assay Kit (Thermo Fisher Scientific, Waltham, MA, USA) as instructed by the manufacturer, and 40 µg of protein were loaded per lane on a Mini-PROTEAN TGX Precast Protein gel (Bio-Rad; Cat. No. 4568094) with the Precision Plus Protein Dual Color Standards Ladder (Bio-Rad; Cat. No. 1610374). Protein

lysates from MCF7 cells were used as positive controls, while protein lysates from HEK293T cells were used as negative controls. Gels were transferred to nitrocellulose membrane using the Turbo-Blot transfer system (Bio-Rad) according to the manufacturer's instructions. The membrane was blocked with 5% milk (Santa Cruz Biotech, Santa Cruz, CA, USA) in TBST 20 mM Tris, pH 7.4 (Lonza, Morrisville, NC, USA), 140 mM NaCl (Lonza), 1% TWEEN-20 (Sigma). PGR protein was detected using Monoclonal Mouse Anti-Human Progesterone receptor clone PgR 1294 (Dako; GA09061-2). B-actin protein was detected using Rabbit polyclonal Actin Antibody (I-19)-R (Santa Cruz Biotech, Dallas, TX, USA; sc-11616-R; Lot#DO406). Both antibodies were diluted 1:1000 in milk and incubated with blots overnight at 4 °C. Bands were detected using Donkey Anti-Mouse (LI-COR Biosciences, Lincoln, NE, USA; 926–32212; Lot # C60524-15) and Goat anti-rabbit (LI-COR Biosciences; 926–32211; Lot # DOO304-15) diluted 1:20,000 in milk and incubated with blots for 45 min at room temperature. Blots were imaged using Odyssey Fc Imager (LI-COR Biosciences) using the 800 nm channel for 10 min and the 700 nm channel for 30 s (to image ladder).

## RNA-Seq library preparation

RNA-Seq libraries were prepared according to the Illumina TruSeq Stranded mRNA protocol Document # 1000000040498 v00. Libraries were sequenced using the Illumina NextSeq 500 and NovaSeq 6000 systems.

## ChIP-Seq library preparation

The ChIP-Seq assay was performed by the Active Motif service laboratory using snap-freeze human myometrial specimens. The H3K27Ac ChIP reactions were conducted with 10 μg of tissue chromatin and 4 μg of H3K27Ac antibody (Active Motif; Cat. No. 39133). The H3K4me1 ChIP reactions were carried out by using 10 μg of tissue chromatin and 4 μl of H3K4me1 antibody (Active Motif, Cat. No. 39297). Libraries were prepared by a custom Illumina library with the standard Illumina PE adaptors (*Short et al., 2008*). Libraries were sequenced using the Illumina NextSeq 500 and NovaSeq 6000 systems.

## Hi-C library preparation

Snap-freeze human myometrial specimens were shipped to the Arima Genomics and the Active Motif service laboratories for preparation of the Hi-C library using the Arima-HiC Kit (Arima Genomics A510008), protocol version A160132 v00. Libraries were sequenced in the NIEHS on the Illumina NovaSeq 6000 platform.

## CRISPR activation (CRISPRa) assay

### Acquisition of guide RNA expression vectors for CRISPRa assay

All gRNA expression vectors were synthesized by and acquired from VectorBuilder. gRNA expressing vectors for PGR targeting (PGR640A and P6SM), non-targeting control, and the empty backbone were purchased from VectorBuilder under the catalog numbers VB191117-1498rhp (PGR640), VB210824-1268wmh (P6SM), VB191117-1500trj, and VB190918-1522gwq, respectively.

### Acquisition of dCas9-VPR-mCherry for CRISPRa assay

IGI-P0492 pHR-dCas9-NLS-VPR-mCherry was a gift from Jacob Corn (Addgene plasmid #102245; http://n2t.net/addgene:102245; RRID:Addgene_102245).

## Production of Lentiviruses

All lentiviruses were packaged in HEK293T/17 cells (ATCC # CRL-11268) according to published method in Current Protocols in Neuroscience (*Chen et al., 2019*). Briefly, 293T cells were transiently transfected with pMD2G, psPAX2, and a transfer vector containing the desired gene using Lipofectamine 2000. Supernatant was collected 48 hr post-transfection and concentrated by centrifugation at 50,000 × *g* for 2 hours. Pellets were resuspended in PBS and used for infection. To determine titer, HEK293T/17 cells were infected with lentiviral samples. Five days post-infection, Qiagen DNeasy Blood & Tissue Kits was used to isolate chromosomal DNA from infected cells. All titers were determined by performing droplet digital PCR (ddPCR) to measure the number of lentiviral particles that integrated into the host genome.

## CRISPRa viral transduction

hTERT-HM cells were infected with lentiviral gRNA expression vectors with a multiplicity of infection of 4 (MOI = 4) and with dCas9-VPR-mCherry at an MOI = 4. gRNA expression vector and dCas9-VPR-mCherry expression vectors, following lentiviral transduction, confer GFP and mCherry fluorescent markers, respectively.

## Fluorescence-activated cell sorting for CRISPRa assay

Cells were examined using a BD FACSAria II cell sorter (Becton Dickinson Biosciences, San Jose, CA) equipped with FACSDiVa software. Initially, a 'scatter' gate was set on a forward scatter (FSC-A) versus side scatter (SSC-A) dot plot to isolate the principal population of cells free of debris. Subsequently, cells were consecutively gated on a side scatter height (SSC-H) versus width (SSC-W), then a forward scatter height (FSC-H) versus width (FSC-W) dot plot to isolate single cells. GFP-expressing cells were excited off a 488 nm laser and detected using a 525/50 nm filter. mCherry-expressing cells were excited off a 561 nm laser and detected using a 610/20 nm filter. Double positive cells were collected based on non-transfected controls and used for further cultural/biochemical analysis.

## Perturb-Seq library preparation

The cells in suspension were counted and examined for viability with trypan blue staining using a TC-20 cell counter (Bio-Rad). Approximately 16,500 live cells at $1 \times 10^6$ cells/ml concentration with 65% or above viability were loaded into the Single Cell Chip to generate single cell emulsion in Chromium Controller with Chromium Single Cell 3' Library & Gel Bead Kit v3.1 (10x Genomics, Cat. 1000268). Reverse transcription of mRNA and cDNA amplification were carried out following the manufacturer's instruction (10 x Genomics, Cat. 1000268, Cat. 1000262 with 10 x Genomics protocol CG000316). The amplified cDNA was separated into CRISPR-sgRNA-derived cDNA and transcriptome-derived cDNA. The CRISPR-sgRNA-derived cDNA was used to make NGS sequencing libraries. The transcriptome-derived cDNA was further fragmented to construct NGS libraries. Both libraries were then sequenced together with the molar ratio of 1–4 by the NIEHS Epigenomics and DNA Sequencing Core Laboratory with the parameters recommended in the manufacturer's instruction.

## Human enhancer perturb-seq data analysis

The raw sequencing FASTQ files generated from both the transcriptome and CRISPR screening libraries were processed together by Cell Ranger software (version 4.0.0, 10 X Genomics). The 'cell-ranger count' pipeline used STAR for aligning the reads to the human reference, GRCh38 'refdata-gex-GRCh38-2020-A' (10 X Genomics), and associated gene expression profile with guide RNA (gRNA) identity by unique barcode in each cell. Seurat software (version 3.6.3) was utilized to perform clustering analysis on the combined dataset (*Satija et al., 2015*). We applied the SCTransform package to normalize gene expression counts across cells (*Hafemeister and Satija, 2019*). The cells were clustered based on the number of unique gRNA types detected. The cell populations containing more than one type of gRNA were excluded from further analyses. We applied two approaches to determine the association between gRNAs and target genes:

A.  For each cell cluster having gRNA with known target genes, such as PLCL1 and PLCL2, we compared the expression of target genes in these clusters to the cluster with scramble control gRNA. We defined individual gRNA activation as a fold change greater than 1.5.

B.  For cell clusters having gRNAs with unknown target genes, we identified differentially expressed genes by performing pair-wise comparisons between each cluster and the cluster with scrambled control gRNA. This analysis was conducted using the 'FindAllMarkers' function in the Seurat package. Target genes were defined as those with a fold change greater than 1.5 and an adjusted p-value less than 0.05. Additional perturb-seq data analysis is provided in *Supplementary file 7*, with a summary table available in *Supplementary file 1*.

## ChIP-Seq

The raw ChIP-seq reads (51 bp, single-end) were filtered with average quality scores greater than 20. Adapter sequences were trimmed from reads using cutadapt (v1.12). Then the reads were aligned to the human reference genome (hg38) using Bowtie (v1.1.2) (*Langmead et al., 2009*), requiring

unique mapping and allowing up to 2 mismatches per read (-m 1 v 2). Duplicated reads with identical sequences were removed using Picard tools. To visualize the read coverage, BigWig files were generated from the bedgraph files of each sample using bedGraphToBigWig. These bigWig files were displayed as custom tracks on the UCSC genome browser.

The normalization of sequencing depth across all ChIP-seq datasets was achieved by downsampling to 20 million uniquely mapped reads per sample. The initial peaks of each sample were identified using MACS2 with a cutoff of adjusted p-value < 0.0001 (*Zhang et al., 2008*). The gene associated with each peak was predicted by searching the TSS of nearby genes within a 100 Kb range using HOMER (*Heinz et al., 2010*). The summary table is available in *Supplementary file 1*.

## Identification of union peaks between H3K27ac and H3K 4me1 peaks

The initial union peaks between H3K27ac and H3K4me1 in each sample were identified using the 'merge' function of BedTools. The common union peaks from multiple samples were used for downstream analysis.

## Identification of union peaks for term pregnant myometrial specimens

Additional H3K27ac ChIP-seq and related input data were obtained from GEO (GSE202027; patients 1, 4, and 7). The same ChIP-seq analysis pipeline described above was applied to those samples. To obtain a comprehensive set of peaks across all H3K27ac ChIP-seq datasets, the peak intervals from each dataset were merged using BedTools.

## Identification of super enhancers

H3K27ac-positive enhancers were defined as regions of H3K27ac ChIP-seq peaks in each sample. The enhancers within 12.5 Kb were merged by using the bedtools merge function with the parameter '-d 12500'. The combined enhancer regions were called super enhancers if they were larger than 15 Kb. The common super enhancers from multiple samples were used for downstream analysis (*Wang et al., 2008*).

Colocalization of super enhancers and PGR genome occupancy was compared by calling peaks from previously published PGR ChIP-seq data (GSM4081683 and GSM4081684). The percentages of enhancers and super enhancers that manifest PGR occupancy were calculated by overlapping the genomic regions in each category with PGR occupancy regions.

## RNA-Seq analysis of term myometrial specimens

The raw paired-end reads (51 bp, paired-end) were initially processed by applying a filter with an average quality score of 20. Adaptor sequences were subsequently removed from reads using cutadapt version 1.12. The processed reads were then aligned to the human reference genome (hg38) using the STAR aligner version 2.5.2b. The gene expression levels in each sample were determined by counting the total number of paired-end reads mapped to each gene using DESeq2 R package version 1.12.4 (*Love et al., 2014*). The gene expression count matrix was normalized based on the ratio of total mapped read pairs in each sample to 56.5 million. Only the genes with average normalized count across all samples larger than one were used for further analysis. The Bioconductor package edgeR was applied to the gene expression count matrix to detect differentially expressed genes between groups of interest (*Robinson et al., 2010*). The false discovery rate (FDR) of differentially expressed genes was estimated using the Benjamini and Hochberg method.

## RNA-Seq analysis of Provera-treated human myometrial specimens

The raw paired-end reads (76 bp, paired-end) were processed by applying a filter with an average quality score of 20. Adapter sequences were removed from reads using cutadapt (v1.12). The processed reads were then aligned to the human reference genome (hg38) using the STAR aligner (v2.5.2b). Fragments Per Kilobase Million (FPKM) was calculated for each gene in each sample by using Cufflinks version 2.0.2.

## Hi-C

The raw reads (51 bp, paired-end) generated from the HiC library were mapped to the human reference genome (hg38; Genome Reference Consortium Human GRCh38 from December 2013) using

HiCUP version 0.7.1 (*Wingett et al., 2015*). The uniquely mapped di-tag passing quality filtering with distance larger than 10 Kb was used for downstream analysis. Chromatin loops were identified using the 'hiccups' function of Juicer version 1.8.9 with default parameters (*Durand et al., 2016*). The A/B compartments were predicted at 100 Kb resolution using the 'eigenvector' function of Juicer. The Hi-C summary table is available in *Supplementary file 1*, and the quality control data are provided in *Supplementary file 2*.

### DNA-binding factor motif enrichment analysis

Enriched motifs were identified by HOMER (Hypergeometric Optimization of Motif EnRichment) v4.11 with default background sequences matching the input sequences (*Heinz et al., 2010*).

### Inferred myometrial PGR activities and the correlation analysis

The inferred PGR activities were represented by the T-score, which was derived by inputting the mouse myometrial *Pgr* gene signature, based on the differentially expressed genes between control and myometrial *Pgr* knockout groups at mid-pregnancy (*Wu et al., 2022*), into the SEMIPs application (*Li et al., 2021*). The T-scores were computed using this signature alongside the normalized gene expression counts (FPKM) from 43 human myometrial biopsy specimens.

## Acknowledgements

We thank the following National Institute of Environmental Health Sciences (NIEHS) Core facilities for exceptional technical support: Epigenomic and DNA Sequencing Core, Viral Vector Core, Flow-cytometry Core, and Integrative Bioinformatics Supportive Group. We thank Ms. Olivia Emery for her technical assistance. This study is partly supported by the Intramural Research Program of the National Institute of Environmental Health Sciences Z1AES103311 (FJD) and by the Gaine Research Foundation GRF-2018–01 (LL). EQ is the awardee of the NIH Intramural Training Program Fellowship. SMR is the recipient of the NIEHS Scholar Connect Program Fellowship.

## Additional information

### Funding

| Funder | Grant reference number | Author |
|---|---|---|
| National Institute of Environmental Health Sciences | Z1AES103311 | Francesco J DeMayo |
| Gaine Research Fund | GRF-2018-01 | Lin Lin |
| National Institutes of Health | NIH Intramural Training Program Fellowship | Elvis Quiroz |
| National Institute of Environmental Health Sciences | NIEHS Scholar Connect Program Fellowship | Skylar G Montague Redecke |

The funders had no role in study design, data collection and interpretation, or the decision to submit the work for publication.

### Author contributions

San Pin Steve Wu, Conceptualization, Formal analysis, Supervision, Writing - original draft, Writing – review and editing; Elvis Quiroz, Formal analysis, Investigation, Methodology, Writing – review and editing; Tianyuan Wang, Data curation, Software, Formal analysis, Writing – review and editing; Skylar G Montague Redecke, Formal analysis, Writing – review and editing; Xin Xu, Data curation, Formal analysis; Lin Lin, Conceptualization, Formal analysis; Matthew L Anderson, Resources, Writing – review and editing; Francesco J DeMayo, Conceptualization, Formal analysis, Supervision, Funding acquisition, Investigation, Project administration, Writing – review and editing

## Author ORCIDs
San Pin Steve Wu ⓘ https://orcid.org/0000-0001-9626-784X
Elvis Quiroz ⓘ https://orcid.org/0009-0000-8702-8054
Tianyuan Wang ⓘ https://orcid.org/0000-0002-3970-0771
Xin Xu ⓘ https://orcid.org/0000-0003-2294-0047
Lin Lin ⓘ https://orcid.org/0000-0002-3228-5739
Matthew L Anderson ⓘ https://orcid.org/0000-0002-2081-4672
Francesco J DeMayo ⓘ https://orcid.org/0000-0002-9480-7336

## Ethics
Human subjects: The deidentified samples were provided from Baylor College of Medicine tissue bank under protocol (H-33461). Informed consent was received from patients which the samples were collected.

Reviewer #1 (Public review): https://doi.org/10.7554/eLife.95897.3.sa1
Reviewer #2 (Public review): https://doi.org/10.7554/eLife.95897.3.sa2
Reviewer #3 (Public review): https://doi.org/10.7554/eLife.95897.3.sa3
Author response https://doi.org/10.7554/eLife.95897.3.sa4

# Additional files

## Supplementary files
Supplementary file 1. Summary table of sequencing datasets.

Supplementary file 2. Summary table of Hi-C quality control metrics.

Supplementary file 3. H3K27ac and H3K4me1 double positive enhancers in term pregnant not in labor human myometrial specimens.

Supplementary file 4. Enrichment of known transcription factor binding motifs in putative myometrial enhancers.

Supplementary file 5. Super enhancers in term pregnant not in labor human myometrial specimens.

Supplementary file 6. Active genes associated with super enhancers in the term nonlabor myometrium.

Supplementary file 7. Cell counts per gRNA and protospacer call frequencies per cell for Perturb-seq analysis.

Supplementary file 8. CRISPRa-dependent gene expression patterns in hTERT-HM cells and gRNA information. DEG, differentially expressed genes between denoted gRNAs.

Supplementary file 9. Normalized gene expression counts across the 43 human myometrial specimens and the PGR T-Scores of individual specimens.

MDAR checklist

## Data availability
Sequencing data have been deposited in GEO under accession codes GSE244735. Materials are available on request by contacting the corresponding author, Francesco DeMayo.

The following dataset was generated:

| Author(s) | Year | Dataset title | Dataset URL | Database and Identifier |
|---|---|---|---|---|
| Wu SP, Quiroz E, Wang T, Redecke SM, Xu X, Lin L, Anderson ML, DeMayo FJ | 2024 | Myometrial specimens | https://www.ncbi.nlm.nih.gov/geo/query/acc.cgi?acc=GSE244735 | NCBI Gene Expression Omnibus, GSE244735 |

The following previously published dataset was used:

| Author(s) | Year | Dataset title | Dataset URL | Database and Identifier |
|---|---|---|---|---|
| Dotts AJ, Reiman D, Yin P, Kujawa S, Grobman WA, Dai Y, Bulun SE | 2022 | In vivo genome-wide PGR binding in pregnant human myometrium identifies potential regulators of labor | https://www.ncbi.nlm.nih.gov/geo/query/acc.cgi?acc=GSE202029 | NCBI Gene Expression Omnibus, GSE202029 |

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
