## [Editor Report · eLife Assessment]

This **valuable** study employed a multi-omics approach to elucidate the regulatory mechanism underlying parturition and myometrial quiescence. The data presented to support the main conclusion are **solid**. This work will be of interest to both basic researchers who work on reproductive biology and clinicians who practice reproductive medicine.

---

## [Referee Report · Reviewer #1 (Public review)]

Summary:

The use of a multi-omics approach to elucidate the regulatory mechanism underlying parturition and myometrial quiescence adds novelty to the study. The identification of myometrial cis-acting elements and their association with gene expression, particularly the regulation of the PLCL2 gene by PGR opens the door to further investigate the impact of PGR and other regulators.

Strengths:

(1) Multi-Omic Approach: The paper employs a comprehensive multi-omic approach, combining ChIP-Seq, RNA-Seq, and CRISPRa-based Perturb-Seq assays, which allow for a thorough investigation of the regulatory mechanisms underlying myometrial gene expression.

(2) Clinical Relevance: Investigating human myometrial specimens provides direct clinical relevance, as understanding the molecular mechanisms governing parturition and myometrial quiescence can have significant implications for the management of pregnancy-related disorders.

(3) Functional work: For functional screening, They have used CRISPRa-based screening of PLCL2 gene regulation using immortalized human cell-line hTERT-HM and T-hESC to add more dimension to the work which strengthens their finding of PGR-dependent regulation of the PLCL2 gene in the human myometrial cells.

Weaknesses:

(1) Variability in epigenomic mapping: The significant variations in the number and location of H3K27ac-positive intervals across different samples and studies suggest potential challenges in accurately mapping the myometrial epigenome. This variability may introduce uncertainty and complicate the interpretation of results.

(2) Sample specificity: The study focuses on term pregnant nonlabor myometrial specimens, limiting the generalizability of the findings to other stages of pregnancy or labor.

(3) Limited Understanding of Regulatory Mechanisms: While the study identifies potential regulatory programs within super-enhancers, the exact mechanisms by which these enhancers regulate gene expression and cellular functions in the myometrium remain unclear. Further mechanistic studies are needed to elucidate these processes.

(4) Discordant analysis: Why regular enhancers are being understood in terms of motif enrichment of transcription factors and super-enhancers in terms of pathways enriched for active genes? This needs a clear reason.

---

## [Referee Report · Reviewer #2 (Public review)]

Summary:

In "Assessment of the Epigenomic Landscape in Human Myometrium at Term Pregnancy" the authors generate a number of genome-wide data sets to investigate epigenomic and transcriptomic regulation of the myometrium at term pregnancy. These data provide a useful resource for further evaluation of gene regulatory mechanisms in the myometrium and include the first Hi-C data published for this tissue. There is a comparison to previously published histone modification data and integration with RNA-seq to highlight potential enhancer-gene regulatory relationships. The authors further investigate putative enhancers upstream of the PLCL2 gene and identify a candidate region that may be regulated by the PGR (progesterone receptor) signaling.

Strengths:

The strengths of this study are in the multi-omics nature of the design as several genome-wide data sets are generated from the same patient samples. Extending this type of approach in the future to a larger number of samples will allow for additional investigation into gene regulation as correlation between epigenomic features and gene expression across a larger number of samples can reveal regulatory relationships.

Weaknesses:

One of the most interesting aspects of this study is the generation of the first Hi-C data for the human pregnant myometrium, however, there is minimal description in the results section of the Hi-C data analysis and the only data shown are the number of loops identified and one such loop that includes the PLCL2 promoter shown in figure 3A. The manuscript would benefit from a more extensive analysis of the Hi-C data, for example, the analysis of TADs (topological associating domains) would be interesting to add and could be used to evaluate to what extent H3K27ac domains and putative regulated genes fall within the same TAD.

The authors present some convincing evidence on the transcriptional regulation of the PLCL2 gene using Perturb-Seq to identify putative upstream enhancer regions and PGR over-expression showing PGR can act as an activator. These two experiments on their own are interesting, however, they are not as mechanistically integrated as they could be to clarify the molecular mechanisms. Deletion of the putative enhancer upstream of PLCL2 followed by over-expression of PGR would clarify the mechanistic relationship between the proposed enhancer, PGR and PLCL2 expression. Does PGR act through the proposed enhancer? In addition, reporter assays using this proposed enhancer region with and without increased expression of PGR and mutation of any PRE sequences would also provide mechanistic insight. Although CRISPRa and Perturb-Seq can be used to identify potential regulatory regions, the best approach to verify the requirement for a particular enhancer in regulating a specific gene is a deletion approach.

Comments on revisions:

The authors have addressed my comments that were directly sent to them, however, my comments in the public review, specifically the superficial nature of the Hi-C analysis were not addressed.

In addition, many of the comments to reviewer 3 were unaddressed and declared out of the scope of this study, as these were points of accuracy in the data analysis they are very much in scope.

I hope the authors reconsider presenting a more thorough analysis.

---

## [Referee Report · Reviewer #3 (Public review)]

In this manuscript, Wu et al. investigate active H3K27ac and H3K4me1 marks in term pregnant nonlabor myometrial biopsies, linking putative enhancers and super enhancers to gene expression levels. Through their findings, they reveal the PGR-dependent regulation of the PLCL2 gene in human myometrial cells via a cis-acting element located 35-kilobases upstream of the PLCL2 gene. By targeting this region using a CRISPR activation system, they were able to increase the elevate the endogenous PLCL2 mRNA levels in immortalized human myometrial cells.

This research offers novel insights into the molecular mechanisms governing gene expression in myometrial tissues, advancing our understanding of pregnancy-related processes.

Major comments:

(1) A more comprehensive analysis of the epigenetic and transcriptomic data would have strengthened the paper, moving beyond basic association studies. Currently, it is challenging to assess the quality and significance of the data as much of the information is lacking.

Strengths:

- The combination of ChIP-Seq, RNA-Seq, and CRISPRa Perturb-Seq approaches to investigate gene regulation and expression in myometrial cells.

- The use of CRISPR activation system to specifically target cis-acting elements.

Weaknesses:

- The manuscript would strongly benefit from a deeper analysis of the Omic datasets. Furthermore, expanding figures/graphs to effectively contextualize these datasets would be greatly beneficial and would add more value to this research.

- Limited sample size, coupled with variability in results and overall lack of details, compromises the robustness of result interpretation.

- Additional efforts are needed to dissect the proposed regulatory mechanisms.

- While the discussion provided helpful context for understanding some of the experiments performed, it lacked interpretation of the results in relation to the existing literature.

Comments on revisions:

The authors have improved the manuscript by enhancing its readability and organization. Tables were added to present key information more clearly, and figures were refined for better visualization. Additionally, more details were included, particularly in the methods and bioinformatics analyses sections, ensuring a more comprehensive and precise presentation of the data.

However, in many cases, reviewers' questions and concerns were addressed in the response to reviewers rather than incorporated into the manuscript, or it was noted that these points would be explored in future studies.

---

## [Author Response]

The following is the authors’ response to the original reviews

**Recommendations for the authors:**

**Reviewer #1 (Recommendations For The Authors):**
(1) Sample size: If the sample size of the study is increased, more confidence and new insights can be inferred about myometrial enhancer-mediated gene regulation in term pregnancy. Such a small sample size (N = 3) limits the statistical power of the study. As mentioned in the manuscript they failed to identify chromatin loops in the second subject's biopsy is observed due to a limited sample.

We agree with the reviewer’s comment about the sample size. We sincerely hope the result of this study would increase the interest of stakeholders to fund future projects in a larger scale.

(2) Figure quality: There is a lack of good representations of the results (e.g., screenshots of tables as figure panels!) as well as missing interpretations that might add value to the manuscript.

Figure 1B and 2B have been converted to the pie chart format.

(3) Definition of super-enhancer: The definition of super-enhancer is not clear. Also, the computational merging of enhancers to define super-enhancers should be described better.

Added more details about tool and parameter setting in the Method section of “Identification of super enhancers”:

“Identification of super enhancers

H3K27ac-positive enhancers were defined as regions of H3K27ac ChIP-seq peaks in each sample. The enhancers within 12.5Kb were merged by using bedtools merge function with parameter “-d 12500”. The combined enhancer regions were called super enhancers if they were larger than 15Kb. The common super enhancers from multiple samples were used for downstream analysis.”

Reference:

Whyte WA, Orlando DA, Hnisz D, Abraham BJ, Lin CY, Kagey MH, Rahl PB, Lee TI, Young RA. Master transcription factors and mediator establish super-enhancers at key cell identity genes. Cell. 2013 Apr 11;153(2):307-19. doi: 10.1016/j.cell.2013.03.035. PMID: 23582322; PMCID: PMC3653129.

(4) Assay-Specific Limitations: Each assay employed in the study, such as ChIP-Seq and CRISPRa-based Perturb-Seq, has its limitations, including potential biases, sensitivity issues, and technical challenges, which could impact the accuracy and reliability of the results. These limitations should be addressed properly to avoid false-positive results and improve the interpretability of the results.

The major limitations of the CRISPRa-based Perturb-Seq protocol in this study are the use of the hTERT-HM cells and the two-vector system for transduction. While hTERT-HM cells are a much easier platform in terms of technical operation, primary human myometrial cells are generally considered retaining a molecular context that is closer to the in vivo tissues. Due to the limitation on the efficiency of having two vectors simultaneously present in the same cell, hTERT-HM cells are much more affordable and operationally feasible to conduct the experiment. Future advancements on the increase of viral vector payload capacity may overcome this challenge and open the venue to perform the assay on primary human myometrial cells.

(5) Sample collection and comparison: There is mention of matched gravid term and non-gravid samples whereas no description or use of control samples was found in the results. Also, the comparison of non-labor samples with labor samples would provide a better understanding of epigenomic and transcriptomic events of myometrium leading to laboring events.

The description has been updated:

“Collection of myometrial specimens

Permission to collect human tissue specimens was prospectively obtained from individuals undergoing hysterectomy or cesarean section for benign clinical indications (H-33461). Gravid myometrial tissue was obtained from the margin of the hysterotomy in women undergoing term cesarean sections (>38 weeks estimated gestational age) without evidence of labor. Non-gravid myometrial tissue was collected from pre-menopausal women undergoing hysterectomy for benign conditions. Specimens from gravid women receiving treatment for pre-eclampsia, eclampsia, pregnancy-related hypertension, or pre-term labor were excluded.”

(6) Lack of clarity:(6a) It is written as 'Chromatin Conformation Capture (Hi-C)'. I think Hi-C is Histone Capture and 3C is Chromosome Conformation Capture! This needs clear writing.

As the reviewer suggested, to make it clear, we have changed the text “A high throughput chromatin conformation capture (Hi-C) assay” to “A High-throughput Chromosome Conformation Capture (Hi-C) assay”.

(6b) In multiple places, 'PLCL2' gene is written as 'PCLC2'.

Corrected as suggested.

(6c) What is the biological relevance of considering 'active' genes with FPKM {greater than or equal to} 1? This needs clarification.

In RNA-seq analysis, the gene expression levels are often quantified using FPKM (Fragments Per Kilobase of transcript per Million mapped reads). Setting a threshold of FPKM for defining "active" genes in RNA-seq analysis is biologically relevant, because it helps to distinguish between genuinely expressed genes and background noise. It helps researchers focus on genes, which are more likely to have a significant biological impact. A common threshold for defining "active" genes is FPKM ≥ 1. Genes with FPKM values below this threshold may be transcribed at very low levels or could be background noise.

(6d) The understanding of differentially methylated genes at promoters is underrated as per the authors. But, why leaving DNA methylation apart, they selected histone modification as the basis of epigenetic reprogramming in terms of myometrium is unclear.

DNA methylation indeed plays a crucial role in evaluating the impact of cis-acting elements on gene regulation. Large-scale studies, such as the comprehensive analysis of the myometrial methylome landscape in human biopsies (Paul et al., JCI Insight, 2022, PMID: 36066972), have provided valuable insights. When integrated with histone modification and chromatin looping data, contributed by our group and collaborators, future secondary analyses leveraging machine learning are poised to further elucidate the mechanisms underlying myometrial transcriptional regulation.

(6e) How does the identification of PGR as an upstream regulator of PLCL2 gene expression in human myometrial cells contribute to our understanding of progesterone signaling in myometrial function?

In a previous study, we demonstrated a positive correlation between *PLCL2* and *PGR* expression in a mouse model and identified PLCL2's role in negatively modulating oxytocin-induced myometrial cell contraction (Peavy et al., PNAS, 2021, PMID: 33707208). The present study builds on this by providing evidence for a direct regulatory mechanism in which PGR influences *PLCL2* transcription, likely through a cis-acting element located 35 kb upstream. These findings suggest that PLCL2 acts as a mediator of PGR-dependent myometrial quiescence prior to labor, rather than merely participating in a parallel pathway. Further in vivo studies are necessary to delineate the extent to which PLCL2 mediates PGR activity, particularly the contraction-dampening function of the PGR-B isoform.

(7) Grammatical error: The manuscript has numerous grammatical errors. Please correct them.

Corrections have been made as suggested.

(8) Use of single-cell data: Though from the Methods section, it can be understood that single-cell RNA-seq was done to identify CRISPRa gRNA expressing cells to characterize the effect of gene activation, some results from single-cell data e.g., cell clustering, cell types, gRNA expression across clusters could be added for better elucidation.

As reviewer suggested, we have prepared a file “PerturbSeq_summary.xlsx” (Dataset S9) to provide additional results of perturb-seq data analysis. It includes 2 spreadsheets, “Cell_per_gRNA” for clustering and “Protospacer_calls_per_cell” for gRNA expression across clusters.

**Reviewer #2 (Recommendations For The Authors):**
(1) The following are a number of grammatical issues in the abstract. I suggest having a careful read of the entire manuscript to identify additional grammatical issues as I may not be able to highlight all of these issues.(1a) "The myometrium plays a critical component during pregnancy." change component to role.(1b) "It is responsible for the uterus' structural integrity and force generation at term," à replace "," with "."(1c) Also, I suggest rephrasing the first 2 sentences to: The myometrium plays a critical role during pregnancy as it is responsible for both the structural integrity of the uterus and force generation at term.(1d) "Here we investigated the human term pregnant nonlabor myometrial biopsies for transcriptome, enhancer histone mark cistrome, and chromatin conformation pattern mapping." Remove "the", and modify to "Here we investigated human term pregnant".(1e) Missing period and sentence fragment, "PGR overexpression facilitated PLCL2 gene expression in myometrial cells Using CRISPR activation the functionality of a PGR putative enhancer 35-kilobases upstream of the contractile-restrictive gene PLCL2.

Corrections have been made as suggested.

(2) Sentence fragment: Studies on the role of steroid hormone receptors in myometrial remodeling have provided evidence that the withdrawal of functional progesterone signaling at term is due to a stoichiometric increase of progesterone receptor (PGR) A to B isoform-related estrogen receptor (ESR) alpha expression activation at term. (Mesiano, Chan et al. 2002) (Merlino, Welsh et al. 2007) (Nadeem, Shynlova et al. 2016).

The statement has been updated:

“Studies on the role of steroid hormone receptors in myometrial remodeling suggest that the withdrawal of functional progesterone signaling at term results from a stoichiometric shift favoring the PGR-A isoform over PGR-B. This shift is associated with increased activation of estrogen receptor alpha (ESR1) expression at term (Mesiano, Chan et al. 2002) (Merlino, Welsh et al. 2007) (Nadeem, Shynlova et al. 2016).”

(3) FOS:JUN heterodimers are implicated to be critical for the initiation of labor through transcriptional regulation of gap junction proteins such as Cx43 (Nadeem, Farine et al. 2018) (Balducci, Risek et al. 1993).Use Gja1 (Gap junction alpha 1) as the current correct gene, not Cx43.Also, several references predate Nadeem, Farine et al. 2018 and are more appropriate to use as references for the role of Ap-1 proteins in regulating Gja1; PMID: 15618352 and PMID: 12064606 were the first to show this relationship in myometrial cells.

The statement has been updated as suggested:

“FOS:JUN heterodimers are implicated to be critical for the initiation of labor through transcriptional regulation of gap junction proteins such as *GJA1* (Nadeem, Farine et al. 2018) (Balducci, Risek et al. 1993)”

(4) Define PLCL2 on first use.

Updated as suggested.

(5) There are a number of issues with this section, "Matched sSpecimens of gravid myometrium were collected at the margin of hysterotomy from women undergoing clinically indicated cesarean section at term (>38 weeks estimated gestation age) without evidence of labor. Specimens of healthy, non-gravid myometrium were also pecimens were collected from uteri removed from pre-menopausal women undergoing hysterectomy for benign clinical indications."

The description has been updated:

“Collection of myometrial specimens

Permission to collect human tissue specimens was prospectively obtained from individuals undergoing hysterectomy or cesarean section for benign clinical indications (H-33461). Gravid myometrial tissue was obtained from the margin of the hysterotomy in women undergoing term cesarean sections (>38 weeks estimated gestational age) without evidence of labor. Non-gravid myometrial tissue was collected from pre-menopausal women undergoing hysterectomy for benign conditions. Specimens from gravid women receiving treatment for pre-eclampsia, eclampsia, pregnancy-related hypertension, or pre-term labor were excluded.”

(6) Enriched motifs were identified by HOMER (Hypergeometric Optimization of Motif EnRichment) v4.11 (Heinz, Benner et al. 2010).

Please clarify what background is used for motif enrichment.

We used the default background sequences generated by HOMER from a set of random genomic sequences matching the input sequences in terms of basic properties, such as GC content and length. We have added more details in the Method section:

“DNA-binding factor motif enrichment analysis

Enriched motifs were identified by HOMER (Hypergeometric Optimization of Motif EnRichment) v4.11 with default background sequences matching the input sequences (Heinz, Benner et al. 2010).”

(7) "Six of the seven regions are also co-localized with previously published genome occupancy of transcription regulators curated by the ReMap Atlas"

Please clarify if this Atlas includes myometrial tissues or not and clarify the cell types included in the atlas.

According to the UCSC Genome Browser and the reference by Hammal et al. (2022), the current ReMap database includes PGR ChIP-seq data from human myometrial biopsies, available under NCBI GEO accession number GSE137550, alongside data from various other cell and tissue types. ReMap provides valuable insights into potential functional cis-acting elements in the genome from a systems biology perspective. However, tissue specificity requires independent validation.

(8) "Notably, 76% of the putative super-enhancers are co-localized with known PGR-occupied regions in the human myometrial tissue (Figure S2). This is significantly higher than the 20% co-localization in the regular enhancer group (Figure S2)."Because there is a huge difference in the size of the putative super enhancer regions and the isolated enhancers this comparison is not appropriate as conducted. The comparison needs to account for the difference in size of the regions. Please provide P values for significance statements.

We acknowledge the reviewer's concern that our initial statement was overstated and potentially misleading, given the substantial difference in size between putative super-enhancer regions and regular enhancers. Rather than emphasizing the enrichment, it would be more accurate to simply describe our observation that super-enhancers encompass more PGR-occupied regions.

Here is the updated version:

“Notably, 76% of the putative super-enhancers co-localize with known PGR-occupied regions in human myometrial tissue, compared to 20% co-localization observed in regular enhancers (Figure S2).”

**Reviewer #3 (Recommendations For The Authors):**
(1) Title is extremely misleading, as here we do not get a view of the epigenomic landscape, but rather sparce data related to H3K27ac and H3K4me (focusing on enhancers) and chromatin conformation associated with the PLCL2 transcription start site (TSS).

As suggested, the title is modified to “Assessment of the Histone Mark-based Epigenomic Landscape in Human Myometrium at Term Pregnancy”.

(2) Improve the first result paragraph by providing a clear rationale for the experiments and their objectives, as well as introducing the samples used. Rather than simply listing approaches and end results in Table 1, offer concise explanations for the experiments alongside the supporting data presented in detailed figures. Using appropriate figures/graphs to effectively contextualize these datasets would be greatly appreciated by readers and would add more value to this research. Currently, it is difficult for us to assess and appreciate the quality of the data.

The following statement is included in the beginning of the Result section:

"To better understand the regulatory network shaping the myometrial transcriptome before labor, we analyzed transcriptome and putative enhancers in individual human myometrial specimens. Using RNA-seq, we identified actively expressed RNAs, while ChIP-seq for H3K27ac and H3K4me1 was used to map putative enhancers. Active genes were associated with nearby putative enhancers based on their genomic proximity. Additionally, chromatin looping patterns were mapped using Hi-C to further link active genes and putative enhancers within the same chromatin loops."

(3) The statistics for every sequencing approach need to be provided for each sample (e.g., RNA-seq: number of total reads, number of mapped reads, % of mapped reads; ChIP-Seq: number of mapped reads, % of mapped reads, % of duplicates).

We have generated the summary table of each dataset included in this study (Dataset S7) [NGS-summary.xls].

(4) Figure S1: The rationale behind comparing the Dotts study and yours regarding H3K27ac-positive regions needs to be better defined. Why is this performed if the data will not be used afterwards? What are the conserved regions associated with vs the ones that are variable? Is this biologically relevant? Why not use only the regions conserved between the 6 samples, to have more robust conclusions?

The purpose of comparing our data with the Dotts dataset is to highlight the degree of variation across studies. In this study, we focused on addressing specific biological questions using our own dataset rather than developing methodologies for meta-analysis. Future advancements in meta-analysis techniques could leverage the combined power of multiple datasets to provide deeper insights.

(5) Perhaps due to a lack of details, I am unable to ascertain how the putative myometrial enhancers were defined. In Dataset S1, it is stated, "we define the regions that have overlapping H3K27ac and H3K4me1 marks as putative myometrial enhancers at the term pregnant nonlabor stage (Dataset S1)". Within Dataset S1, for subjects 1, 2, and 3, H3K27ac and H3K4me1 double-positive enhancers are shown in term pregnant, non-labor human myometrial specimens, with approximately 100 regions corresponding to 131 (sample 1), 127 (sample 2), and 140 (sample 3) common peaks. However, in Figure 1a, reference is made to the 13114 putative enhancers commonly present across the three specimens. Is Dataset S1 intended to represent only a small fraction of the 13114 putative enhancers? Detailed analyses need to be conducted and better showcased.

Dataset S1 has been updated to list all 13,114 putative enhancers.

(6) For the gene expression analyses of RNA-seq data, FPKM values were utilized. However, it is unclear why the gene expression count matrix was normalized based on the ratio of total mapped read pairs in each sample to 56.5 million for the term myometrial specimens. I would recommend exercising caution regarding the use of FPKM expression units, as samples are normalized only within themselves, lacking cross-sample normalization. Consequently, due to external factors unaccounted for by this normalization method, a value of 10 in one sample may not equate to 10 in another.

We value the reviewer’s input. This question will be addressed in future secondary data analyses with suitable methodologies, as it is beyond the scope of this study.

(7) In Figure 1b, the authors have categorized their 12157 active genes into 3 bins based on FPKM values: >5 FPKM >1, >15 FPKM >5, and >15 FPKM. However, in the text, they describe these as 'actively high-expressing genes (FPKM >= 15)'. I would advise caution regarding the interpretation of these values, as an FPKM of 15 is not typically associated with highly expressed genes. According to literature and resources such as the Expression Atlas, an FPKM of 15 is generally considered to represent a low to medium expression level.

We appreciate the reviewer’s feedback. This question will be revisited during secondary data analyses using appropriate methodologies, as it falls outside the scope of the present study.

To increase readability and clarity, we modified the sentence as following: More than 40% of the 540 putative super enhancers are located within a 100-kilobase distance to high-expressing genes (FPKM >= 15), while only 7.3% of putative myometrial super enhancers are found near low-expressing genes (5 > FPKM >= 1) (Figure 2B).

(8) Out of the 12157 active genes, approximately two-thirds have an FPKM >15. Was this expected? How does this correspond to what is observed in the literature, particularly in other similar studies (https://pubmed.ncbi.nlm.nih.gov/30988671/ ; https://pubmed.ncbi.nlm.nih.gov/35260533/).

This is indeed an intriguing question that merits further exploration in future secondary analyses.

(9) It is also surprising to see that for the motif enrichment analysis (Fig. 1C), the P-values are small. This is probably because the percentage of target sequences with the motif is very similar to the percentage of background sequences with the motif. For instance, for selected genes in Figure 1C: AP-1 (50.68% vs. 46.50%), STAT5 (28.08% vs. 25.04%), PGR (17.90% vs. 16.12%), etc. Can one really say that you have a biologically relevant enrichment for values that are so close between target sequences and background sequences?

Reviewer’s comment is noted. Biological relevance shall be experimentally examined though wet-lab assays in future studies.

(10) For Figure 2, again not convinced that FPKM >= 15 can be used to say: Compared with the regular putative enhancers, the putative myometrial super-enhancers are found more frequently near active genes that are expressed at relatively higher levels (Figure 1B and Figure 2B). A higher threshold should be used if they want to say this.

To compare the association of putative enhancers with active genes expressed at different levels, we categorized the active genes into three groups based on their FPKM (Fragments Per Kilobase of transcript per Million mapped reads) values. These groups are defined as follows: the top third active genes (FPKM ≥ 15), the middle third active genes (5 ≤ FPKM < 15), and the bottom third active genes (1 ≤ FPKM < 5). By "active genes expressed at relatively higher levels," we refer specifically to the top third active genes with FPKM values of 15 or higher, indicating their relatively higher expression levels compared to the other groups of active genes.

(11) More detailed explanations and methods are needed regarding how the data for Figure S2 was obtained.

The following details were added to the methods section:

“Colocalization of super enhancers and PGR genome occupancy was compared by calling peaks from previously published PGR ChIP-seq data (GSM4081683 and GSM4081684). The percentages of enhancers and super enhancers that manifest PGR occupancy were calculated by overlapping the genomic regions in each category with PGR occupancy regions.”

(12) In Figure 2C, there is no information provided on the genes used to obtain the results. It would be helpful to include examples of these genes, along with their expression values, for instance.

The expression levels of the 346 active genes that are associated with myometrial super enhancers are included in Dataset S4, along with results of the updated gene ontology enrichment analysis using the Database for Annotation, Visualization, and Integrated Discovery (DAVID) of Knowledgebase v2024q4. Selected pathways of interest are listed in updated Figure 2C.

(13) The linking of PLCL2-related data to the first part of the story is lacking, and the rationale behind it is missing. This entire section should be more detailed, and the data should be expanded to better reflect the context.

As suggested, we included the following statement at the beginning of the section “Cis-acting elements for the control of the contractile gene PLCL2”:

“We previously demonstrated the positive correlation of PLCL2 and PGR expression in a mouse model and PLCL2’s function on negatively modulating oxytocin-induced myometrial cell contraction (Peavy et al., 2021). However, the mechanism underlies the PGR regulation of PLCL2 remains unclear. Taking advantage of the mapped myometrial cis-acting elements, we aimed to identify the cis-acting elements that may contribute to the PLCL2 transcriptional regulation with a special interest on the PGR-related enhancers.”

The context is that our results provide additional evidence to support a direct regulation mechanism of PGR on the *PLCL2* transcription, likely though the 35-kb upstream cis-acting element. This finding suggests that PLCL2 likely plays a mediator’s role of PGR dependent myometrial quiescence before laboring rather than a mere passenger on a parallel pathway. Further studies using in vivo models are needed to determine the extent of PLCL2 in mediating PGR, especially PGR-B isoform’s contraction-dampening function.

(14) The entire Hi-C data should be presented to allow for the assessment of its quality and further value.

The revised manuscript has included the Hi-C quality control summary in Dataset S8 [HiC-QC-Summary.xlsx].

(15) The authors state: "For the purpose of functional screening, we focus on H3K27ac signals instead of using H3K27ac/H3K4me1 double positive criterium to cast a wider net." However, it is unclear how many of the targeted regions contained H3K27ac/H3K4me1 peaks. Were enhancers or super-enhancers targeted, and if so, how did they compare to H3K27ac sites?

The numbers of H3K27ac/H3K4me1 double positive peaks are recorded in Figure 1A. Compared to the numbers of H3K27ac intervals (Table 1), the H3K27ac/H3K4me1 double positive peaks are 62.9%, 70.7%, and 61.2% of corresponding H3K27ac intervals in each individual specimen.

(16) For the first set of data (Table 1), the authors state, "Together, these results reveal an epigenomic landscape in the human term pregnant myometrial tissue before the onset of labor, which we use as a resource to investigate the molecular mechanisms that prepare the myometrium for subsequent parturition." While it is acknowledged that an epigenetic landscape exists in all tissues, there is a lack of clarity regarding this landscape in the current manuscript, as we are only presented with a table containing numbers.

This sentence has been revised to: “Together, these results delineate a map of H3K27ac and H3K4me1 positive signals in the human term pregnant myometrial tissue before the onset of labor, which we use as a resource to investigate the molecular mechanisms that prepare the myometrium for subsequent parturition.”

(17) For S1, the authors conclude: These data together highlight the degree of variation in mapping the epigenome among specimens and datasets. This conclusion seems somewhat perplexing, and I find myself in partial disagreement. Firstly, providing a clear rationale for this section would strengthen the conclusions. It's important to consider what factors may contribute to this variability. It could simply be attributed to differences in experimental settings, such as variations in samples, protocols used, antibodies, sequencing departments, or overall data quality. Deeper analyses of the data could have provided more information.

We agree with the reviewer that deeper analyses are needed in order to extract more information among studies. However, appropriate methods for meta-analyses should be carefully evaluated and employed for this purpose. We humbly believe that such a task should belong to future studies that may combine available datasets for secondary analyses, leveraging the collective contribution of the reproductive biology community.

(18) In the methods section, please include an explanation of how enhancers and super-enhancers were defined or add appropriate citations for reference.

Added more details about tool and parameter setting in the Method section of “Identification of super enhancers”.

“Identification of super enhancers

H3K27ac-positive enhancers were defined as regions of H3K27ac ChIP-seq peaks in each sample. The enhancers within 12.5Kb were merged by using bedtools merge function with parameter “-d 12500”. The combined enhancer regions were called super enhancers if they were larger than 15Kb. The common super enhancers from multiple samples were used for downstream analysis.”

Reference:

Whyte WA, Orlando DA, Hnisz D, Abraham BJ, Lin CY, Kagey MH, Rahl PB, Lee TI, Young RA. Master transcription factors and mediator establish super-enhancers at key cell identity genes. Cell. 2013 Apr 11;153(2):307-19. doi: 10.1016/j.cell.2013.03.035. PMID: 23582322; PMCID: PMC3653129.

(19) Additional description on the "Inferred myometrial PGR activities and the correlation analysis "method section should be included to enhance clarity and understanding.

The description has been updated:

“The inferred PGR activities were represented by the T-score, which was derived by inputting the mouse myometrial *Pgr* gene signature, based on the differentially expressed genes between control and myometrial *Pgr* knockout groups at mid-pregnancy (Wu, Wang et al., 2022), into the SEMIPs application (Li, Bushel et al., 2021). The T-scores were computed using this signature alongside the normalized gene expression counts (FPKM) from 43 human myometrial biopsy specimens.”

(20) How was the qPCR analysis performed? Was the ddCT method utilized, and was a reference gene used for control? Additional information would be beneficial.

Quantifying relative mRNA levels was performed via the standard curve method.

The following details were added: “Relative levels of genes of interest were normalized to the 18S rRNA.”

(21) Regarding the RNA-Seq analysis of Provera-treated human Myometrial Specimens, the continued use of FPKM is not ideal due to potential differences in RNA composition between libraries. Additionally, clarification is needed on why Cufflinks 2.0.2 was used, considering it is no longer supported.

FPKM (Fragments Per Kilobase of transcript per Million mapped reads) is used in RNA-Seq analysis, because it allows for the normalization of gene expression data, accounting for differences in gene length and sequencing depth, and facilitates comparability across different genes and libraries. This makes it one of the essential tools for accurately measuring and comparing gene expression levels in various biological and clinical research contexts.

CuffLinks was once a popular tool for analyzing RNA-seq data, transcriptome assembly, and DEG identification. Its usage has declined in recent years due to the emergence of newer and more advanced tools. The main reason is that it was used for RNA-seq analysis at early stage of this study a few years ago. For the purpose of comparison and consistency, we continued using this tool for later RNA-seq analysis. If we start a new project now, we will choose newer tools, such as HISAT2, Salmon, and DEseq2.

(22) Overall, sentence structure and typos need to be corrected across the text. Here are some examples:Line 17: at term, emerging studies.Line 20-22: Here we investigated the human term pregnant nonlabor myometrial biopsies for transcriptome, enhancer histone mark cistrome, and chromatin conformation pattern mapping.Line 30-32: PGR overexpression facilitated PLCL2 gene expression in myometrial cells Using CRISPR activation the functionality of a PGR putative enhancer 35-kilobases upstream of the contractile-restrictive gene PLCL2.Line 66-70: However, the role of differential myometrial DNA methylation at contractility-driving gene promoter CpG islands in preterm birth is not thought to be major (Mitsuya, Singh et al. 2014), but given that DNA methylation-mediated gene regulation often occurs outside of CpG islands (Irizarry, Ladd-Acosta et al. 2009), there is still work to be done at this interface.Line 80-83: Putative enhancers upstream of the PLCL2, a gene encoding for the protein PLCL2 which has been implicated in the modulation of calcium signaling (Uji, Matsuda et al. 2002) and maintenance of myometrial quiescence (Peavey, Wu et al. 2021), transcriptional start site were subject to functional assessment using CRISPR activation based assays.Line 290 : sSpecimens

We appreciate the reviewer’s kind efforts and have made changes accordingly.